

# First report of microplastics in the gastrointestinal tracts of North American insectivorous bats

Ashleigh B. Cable[1], Emma V. Willcox[1], Leah N. Crowley[1] and Christy Leppanen[2]

[1] School of Natural Resources, University of Tennessee - Knoxville, Knoxville, TN, United States of America
[2] The Center for Tobacco Products, United States Food and Drug Administration, Silver Spring, MD, United States of America

## ABSTRACT

**Background**. Microplastics (MPs) are among the many ubiquitous environmental contaminants of emerging concern for both aquatic and terrestrial species. Bats have integral roles in aquatic-terrestrial food webs on almost every continent, are exposed to a wide variety of environmental contaminants, and yet have received limited investigation about the threat of MPs. While MPs have been detected in numerous bird species and in bats of the Amazon, there are no published studies documenting the dietary MP exposure of North American bats that consume many terrestrial and aquatic arthropods or the possible adverse effects of exposure.

**Methods**. We chemically digested bat gastrointestinal tracts (GITs) to extract, quantify, and characterize MPs that accumulated in insectivorous *Eptesicus fuscus* (big brown bats). We quantified MPs in procedural blanks to account for background contamination in the lab for controls and compared concentrations in bat GITs to controls. We measured the mass of bat carcasses, minus the brains, prior to necropsy to determine body condition. We investigated the relationship between MP concentrations in bat GITs to body condition to determine if higher concentrations indicated reduced bat body condition using linear regression.

**Results**. Our results indicate that the ingestion of MPs by bats could lead to lower bat mass, potentially related to poorer body condition or ability to store fat. The ability to store and use fat is crucial for the survival of these migrating and cave-hibernating species. Moreover, bats with higher fat stores are more likely to survive multiple stressors such as the non-native fungal disease white-nose syndrome. This study will allow future research to build off baseline information and further explore the effects of MPs to individuals and populations of bats of conservation concern.

## INTRODUCTION

Microplastics (MPs), or tiny plastic particles sized 1 μm–5 mm, are human-sourced contaminants that are widespread and abundant in the environment (*Hale et al., 2020*; *Thacharodi et al., 2024*). Nano- (<1 μm) and meso- (5–25 mm) sized plastic particles are often grouped together with MPs in research studies (*Correia et al., 2023*). Considerable

Corresponding author
Ashleigh B. Cable,
acable5@vols.utk.edu

research on MPs has focused on uptake by aquatic organisms, but recent studies also show that terrestrial organisms are exposed (*Ayala et al., 2023*; *Carlin et al., 2020*; *Fackelmann et al., 2023*; *Hoang & Mitten, 2022*; *Masiá, Ardura & Garcia-Vazquez, 2019*; *Sherlock et al., 2022*; *Wayman et al., 2024*; *Weitzel et al., 2021*; *Winkler et al., 2020*). Potential health concerns related to MP exposure are numerous across taxa and include physiological reactions to the constituents from which they are made, such as bisphenol A (BPA), a known endocrine-disrupting compound (*Flint et al., 2012*). Emerging research indicates that the physical properties of MPs are implicated in the disease plasticosis, diagnosed by particles that embed in or change the structure of tissues (*Charlton-Howard et al., 2023*). Microplastics have been associated with many sublethal effects in organisms including reduced body condition (*Welden & Cowie, 2016*), altered gut microbiomes (*Fackelmann et al., 2023*), altered fatty acid composition (*McCann Smith et al., 2024*), organ damage (*Rivers-Auty et al., 2023*), depressed immune systems, oxidative stress, and inhibited growth (*Osman et al., 2023*).

Aerial insectivores are among the terrestrial organisms that are exposed to MPs. In the last five years, numerous studies have documented MP exposure in birds (*Carlin et al., 2020*; *Fackelmann et al., 2023*; *Liu et al., 2023*; *Masiá, Ardura & Garcia-Vazquez, 2019*; *Schutten et al., 2024*; *Tatlı et al., 2025*; *Teboul et al., 2021*; *Tokunaga et al., 2023*; *Wayman et al., 2024*). One published study documents exposure in multiple bat species in the Brazilian Amazon (*Correia et al., 2023*). To date, no studies in the literature have investigated MP exposure in North America (NA) insectivorous bats. Many of these bat species are of conservation concern due to a myriad of threats, including habitat loss or degradation, urbanization, agricultural intensification, wind turbine fatalities, and the disease white-nose syndrome (WNS) (*Cable, Willcox & Leppanen, 2022*; *Frick, Kingston & Flanders, 2020*; *Hoyt, Kilpatrick & Langwig, 2021*; *Korine et al., 2016*; *Kunz et al., 2007*; *Put, Fahrig & Mitchell, 2019*). The objectives of this study were to (1) test a method for extracting microplastics from bat stomach contents, (2) characterize and quantify MP concentrations in bat gastrointestinal tracts, and (3) investigate the possible relationship of MP concentrations with bat sex, age, and body condition.

## MATERIALS & METHODS

### Collection of samples for protocol testing

We received bat stomach contents collected during previous field studies as part of WNS surveillance. At the time of collection, WNS was recently introduced to NA, and large quantities of dead bats were found in caves. The samples used in this study were collected from three locations in the Northeast USA: Bennington County cave, Vermont ($n = 42$), Hampden County mine, Massachusetts ($n = 25$), and Warren County mine, New York, USA ($n = 25$). We assume that they were all collected from sites by researchers and know that they were all from a single species, *Myotis lucifugus* (little brown bat). The samples were not collected for the purpose of studying microplastics and there was no protocol in place to limit contamination at time of stomach content collection; therefore, we only used these samples to modify a method of MP extraction and quantification and cannot determine the source of MPs in these samples.

## Bat gastrointestinal tract (GIT) collection

We collected bat carcasses through public health monitoring programs in Tennessee, USA. We received dead bats that had been collected throughout the state and submitted to facilities in Knox and Davidson Counties. Bats were often submitted following human or pet exposure and all were missing brain tissue from rabies tests that were performed. None of these bats were tested for the presence of WNS. We chose *Eptesicus fuscus* (big brown bat) as our study species because we had the highest number of carcasses of that species. Bats were frozen at $-20\,°C$, then thawed for three hours on the day of processing and necropsied in a dedicated necropsy lab at the University of Tennessee College of Veterinary Medicine to extract the full, intact gastrointestinal tracts (GIT) from esophagus to anus. During necropsy, we placed bats on metal pans, necropsied them with small metal scissors and scalpels, and stored all extracted organs in aluminum foil. We did not rinse equipment during necropsy but kept organs intact to avoid contamination of the internal organs. External contamination of organs was addressed in the microplastic lab on the day of digestion (see below). We refroze each sample in labeled whirl pak bags until further analysis. Unlike the collection method of the stomach content samples discussed previously, this GIT collection protocol ensured reduced risk of MP contamination, thus allowing us to test the dietary pathway of exposure.

## Extraction of microplastics

We adapted methods previously used to extract MPs from GITs of birds of prey (*Carlin et al., 2020*). We rinsed each intact GIT sample with pure deionized water (DI water) twice to remove external microplastics. This step was not conducted with the stomach contents samples as that would have washed away all the material. A solution of potassium hydroxide (KOH) was prepared from pellets (Macron Fine Chemicals, Batch No. 0000271176) and deionized water so that the concentration was 10% KOH. The KOH was not filtered so as not to introduce MPs from the container that it would be filtered into in an additional step. Instead, the solution was used in both tissue digestion and the processing of blanks to quantify contamination (see below). We weighed each GIT sample and digested tissues in the KOH solution and placed them on a shaker at 180 revolutions per minute (RPM) speed for 24 h. The amount of KOH used for each sample was determined by using an analytical scale to weigh the GIT sample and aimed to add 3 times the weight in KOH. All GITs were fully submerged in KOH despite the small volumes of solutions used. Samples remained in the KOH solution for 3–12 days depending on tissue digestion success. While *Carlin et al. (2020)* applied heat to the digestion of GITs of large birds of prey, we found during protocol testing that applying heat during digestion was not required, possibly due to the small sample mass of bat GITs. We used a glass vacuum filter system and glass membrane filters (Whatman, grade GF/F borosilicate glass microfiber filters, 47 mm diameter, 0.42 mm thickness) to separate out undigested from digested material. We originally used filters with a 2.7 $\mu$m pore size for stomach content samples but determined that a smaller pore size could be used compared to those used for large birds (*Carlin et al., 2020*) so we modified the protocol to use 0.7 $\mu$m pore size for GIT samples. We stored
samples on the glass filters in closed glass petri dishes in a dark filing cabinet until we counted particles in subsequent steps.

## Limiting and quantifying contamination

To avoid introducing contamination to samples, all benches were wiped down at the beginning of each session with paper towels and DI water. We wore 100% white cotton lab coats, covered all samples with aluminum foil while processing, and rinsed all glassware with pure DI water. We incorporated two procedural blanks every digestion session in the lab to quantify background contamination following the same procedures that we used to digest tissue samples (rinsing glassware, adding 10% KOH, filtering through vacuum filter, storing filters in petri dishes, and counting and characterizing particles). The procedural blanks only contained the average amount of KOH solution (in grams using an analytical scale) that was added to the previous ten GIT samples and did not contain any tissues. We attempted to limit the number of people in the lab, but it was a shared space, and we could not always control lab use.

## Counting and characterizing microplastics

We examined each glass filter under a microscope (Motic, Model SMZ-171) and scanned left-to-right and up-to-down. We scanned on multiple magnification settings ranging 7.5–50X to count particles of various sizes. We placed a dot, made with a fine tipped marker, beside every piece of plastic to avoid counting particles more than once. We characterized each particle type based on the type and color (*Carlin et al., 2020*; *Rodríguez-Seijo & Pereira, 2017*). We used a hot needle method to determine if some particles were plastics. First, we would heat the needle with a flame and place it on the edge of a particle. If it melted or bent with heat, we classified it as an MP. If it did not melt or bend, we applied pressure to the particle with the needle. If the particle broke under pressure, we did not classify the particle as an MP. Chitinous insect parts that remained post-digestion were not classified as MPs. They did not melt or bend with heat, and they would often shatter under needle pressure. Although the hot needle method is sometimes considered outdated now, it has been used for nearly a decade and was a common method at the time of this study (*De Witte et al., 2014*). If we were not sure about a particle, we were conservative and did not classify it as an MP. We also recorded the color of each particle, as color is often characterized in MP research (*Carlin et al., 2020*; *Masiá, Ardura & Garcia-Vazquez, 2019*). We used a Moticam X3 camera and software to take pictures and measure the longest side of a subset of particles per sample.

## Microplastics and age, sex, and body condition

We calculated MP concentration by taking the number of particles divided by the mass of the GIT tissue plus the added KOH solution. Microplastic concentration for procedural blanks (*i.e.,* controls) were the number of particles divided by the mass of the KOH solution only. We log transformed MP concentrations from GITs to normalize the data for use in the statistical analysis but report raw concentrations in the results. We used analysis of variance (ANOVA) to test if MP concentrations in bat samples were significantly different than control samples. We used ANOVA to also determine if MP concentrations varied

**Table 1 Summary table of microplastics (MPs) found in the gastrointestinal tracts (GITs) of big brown bats (*Eptesicus fuscus*).** Samples were collected from 15 counties in Tennessee, United States of America in years 2019 and 2020, and procedural blanks (*i.e.*, control samples). Concentrations are reported as number of MPs divided by the mass of the tissue sample (g) plus KOH solution (g) for the GITs and number of MPs divided by the KOH solution (g) for procedural blanks.

| Sample type | MP count (*n*) (mean ± SD) | Total count (*n*) (range) | MP concentration (*n*/gram) (mean ± SD) | Total MP concentration (*n*/gram) (range) |
|---|---|---|---|---|
| GITs (*n* = 26) | 22.1 ± 25.6 | 574 (1–112) | 7.5 ± 10.9 | 194.3 (0.3–54.6) |
| Controls (*n* = 10) | 2.8 ± 1.7 | 28 (1–6) | 1.2 ± 0.5 | 11.6 (0.4–2.0) |

by age and sex. We graphed body mass and MP concentrations by collection month to investigate seasonal patterns of MP concentrations and bat mass. We fit a linear regression model in R version 4.3.2 to determine if MP GIT concentration had a relationship with bat mass, a proxy for estimating fat stores (*McGuire et al., 2018*). Bat mass was the dependent variable and MP concentration was the independent variable.

## RESULTS

### Protocol testing with bat stomach contents

Methods used previously for extracting MPs from GITs of birds of prey (*Carlin et al., 2020*) and adapted for this study extracted 306 MPs from 85 of the 92 *M. lucifugus* stomach contents. Fibers made up over half of the observed plastics (56%), followed by fragments (35%), films (4%), foams (3%), and fiber bundles (2%). Blue particles made up over half of the observed plastics (56%), followed by clear (16%), and red particles (12%). The remaining 16% included black, brown, gray, green, pink, purple, white, and yellow particles. This determined that the method was appropriate for extracting different particle types from bat samples, that applying heat during the digestion was not necessary, and that a smaller pore size filter could be used in subsequent steps with GITs.

### Microplastics concentrations and characterizations

We extracted 608 plastic particles from 26 *E. fuscus* GITs collected from 15 Tennessee counties (Fig. 1) and 28 plastic particles from 10 control samples. The average MP concentration in GITs was 7.5 *n*/g ± 10.9 SD (Table 1). The majority of microplastics were fibers (*n* = 574; 94%; Fig. 2). We also identified nine fiber bundles, 16 fragments, seven films, and two spheres. We identified 26 fibers and two foams in our 10 control samples. Control samples had an average MP concentration of 1.2 *n*/g ± 0.5 SD (Table 1). All plastics in control samples were clear fibers (64%), blue fibers (18%), purple fibers (11%), or blue foams (7%). The most abundant MPs in GIT samples were clear fibers (52%), blue fibers (35%), red fibers (8%), purple fibers (2%), blue fragments (1%), clear fiber bundles (1%), and white films (1%). We also identified fibers of other colors and other plastics occurring in smaller numbers. To be conservative in our estimates of MPs, we only used fibers in the statistics because they were the easiest to tell apart from natural material and were the most frequent type occurring in bat samples.

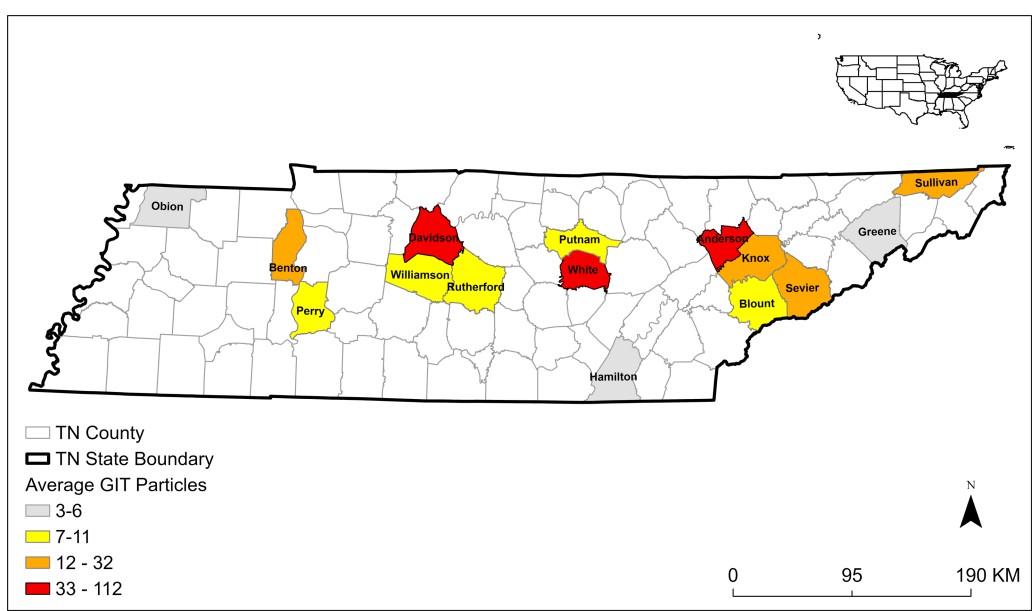

**Figure 1** Map of 15 Tennessee (TN) counties and the average number of microplastics extracted from gastrointestinal tracts (GITs) from bats collected in 2019 and 2020.

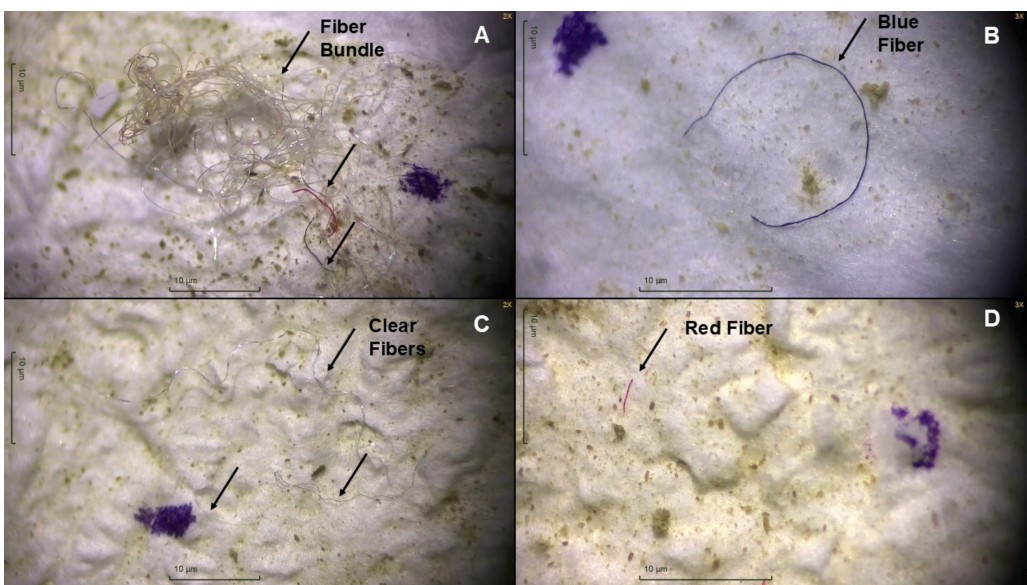

**Figure 2** Microscopic images of plastic particles extracted from gastrointestinal tracts of big brown bats (*Eptesicus fuscus*) collected in 2019 and 2020 in Tennessee, USA. (A) Fiber bundle with clear, red, and blue fibers, (B) blue fiber, (C) three clear fibers, and (D) a red fiber extracted from bat gastrointestinal tracts. Each black arrow points to a fiber. The purple marker used when counting plastics is visible in all four photos. Photos were taken with Motic Images Plus 3.0.

**Table 2  Mean and standard deviation values for microplastic (MP) concentrations (number of MPs divided by sample and KOH mass (g)) in big brown bat (*Eptesicus fuscus*) gastrointestinal tracts.** Samples were collected from 15 counties in Tennessee, United States in years 2019 and 2020. The number in parentheses indicates sample size.

| Sex | Juvenile | Adult | Unknown | All |
|---|---|---|---|---|
| Male | 3.5 ± 0.8 (3) | 13.1 ± 16.0 (10) | | 10.9 ± 14.5 (13) |
| Female | 6.7 ± 5.7 (4) | 2.9 ± 2.6 (8) | 2.6 (1) | 4.0 ± 3.9 (13) |
| All | 5.3 ± 4.4 (7) | 8.6 ± 12.8 (18) | 2.6 (1) | |

## Microplastics and age, sex, and body condition

Microplastic concentrations were significantly higher in GIT samples than control samples (Table 1; $p = 0.01$). Concentrations were not significantly higher in males ($n = 13$) than females ($n = 13$) in the ANOVA analysis ($p = 0.08$). Higher concentrations in adults ($n = 18$) compared with juveniles ($n = 7$) were also not significant (Table 2; $p = 0.96$). The linear model determined that MP fiber concentrations in the GIT had a significant negative association with bat body condition, with lower body mass associated with higher GIT MP concentrations (Fig. 3; $p = 0.007$, adjusted $r^2 = 0.23$, $F_{1,24} = 0.72$ [Bat mass $= 15.77 - 1.37$ * log (MP concentration)]).

*Eptesicus fuscus* were collected in April ($n = 1$), May ($n = 8$), June ($n = 4$), July ($n = 5$), August ($n = 3$), September ($n = 1$), and November ($n = 4$). We did not compare body mass and MP concentrations across months statistically due to limited sample size. Mean bat body mass appeared similar in all months (12.4–14.1 g range) except November (17.5 g ± 3.7 SD; Fig. 4). Microplastic concentrations appeared lowest in April (0.7 $n$/g), but the sample size was a single bat. May appeared to have the highest MP concentrations (11.7 $n$/g ± 18.3 SD; Fig. 4).

## DISCUSSION

Emerging research on MPs has demonstrated that they are ubiquitous environmental contaminants. We show in this study that insectivorous bats are no exception to the terrestrial wildlife species that ingest MPs. To our knowledge, this is the first study to document dietary exposure routes of MPs to bats in North America; however, other studies have similar findings in 25 bat species captured in the Brazilian Amazon (*Correia et al., 2023*) and it is likely a global issue. The MP concentrations that we detected in bat GITs were comparable to GITs of migratory birds and nestling birds in Wisconsin and Illinois, USA (*Hoang & Mitten, 2022*). The presence of MPs in GITs suggests that bats are exposed from their insectivorous prey, by contaminated drinking water, or by incidental or intentional ingestion of particles suspended in the air. All are possible explanations as ontogenic transfer of MPs from aquatic-to-terrestrial systems through metamorphosis has been documented in arthropods (*Al-Jaibachi, Cuthbert & Callaghan, 2019*; *Grgić et al., 2023*; *Yıldız et al., 2022*). Considering that bats are known to consume a variety of arthropods (*Deeley et al., 2023*; *Maslo et al., 2022*), exposure *via* prey items is possible. Moreover, environmental MPs in freshwater systems and the air are abundant and widespread, thus available for uptake (*O'Brien et al., 2023*; *Thacharodi et al., 2024*).

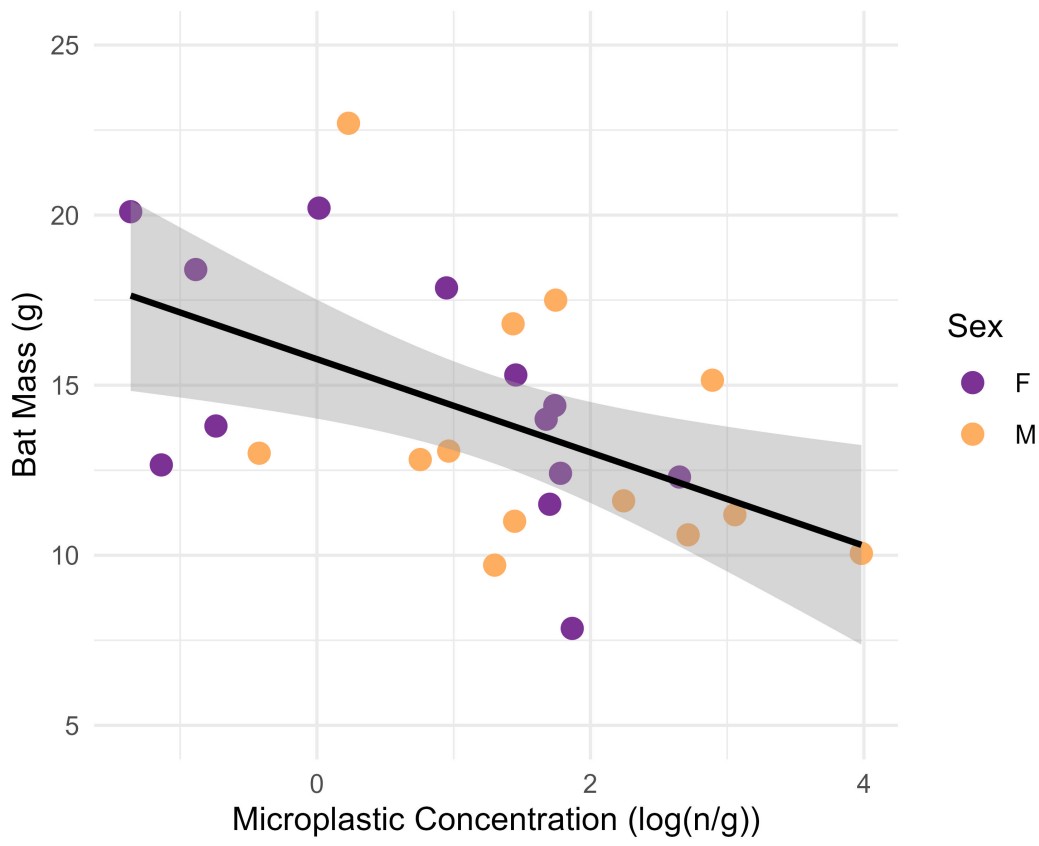

**Figure 3** **Plot showing the linear relationship between the log-transformed microplastics concentrations (n particles/gram tissue) in the gastrointestinal tracts of big brown bats (*Eptesicus fuscus*) and bat mass of whole carcass bat minus brains (grams).** Samples were collected from 15 counties in Tennessee, USA in years 2019 and 2020. The line shows the fit linear model (Body mass = 15.77–1.37 (log (MP concentration)); $F_{1,24} = 8.72$, $p = 0.007$, adjusted $r^2 = 0.23$) and the gray shading is the 95% confidence interval. There is a significant relationship between microplastics concentrations and bat mass.

Bats visit water sources for foraging and drinking purposes (*Kalcounis-Rueppell et al., 2007*; *Rydell et al., 2022*), thus, drinking water could be a route of exposure. Bats might also ingest microplastics during grooming as they can with other contaminants (*Pitt et al., 2014*; *Racey & Swift, 1986*; *Shore et al., 1991*). Microplastic pollution in the air might also have substantial influence on bat exposure, as MPs are abundant in the atmosphere (*O'Brien et al., 2023*) and bats use the aerosphere to forage. Future research could explore MP concentrations in arthropods that are known bat prey sources, surface water and air from foraging and drinking areas, and concentrations in bat organs and bat guano to understand accumulation patterns. The effects of MP air pollution on bats could be important to understand, especially if bats confuse plastics for prey items or if the suspended particles interfere with bat echolocation ability. Similar misperceptions of acoustic signals associated with marine debris that are hypothesized to influence foraging by echolocating cetaceans (*Merrill et al., 2024*) might apply to aerial echolocating species.

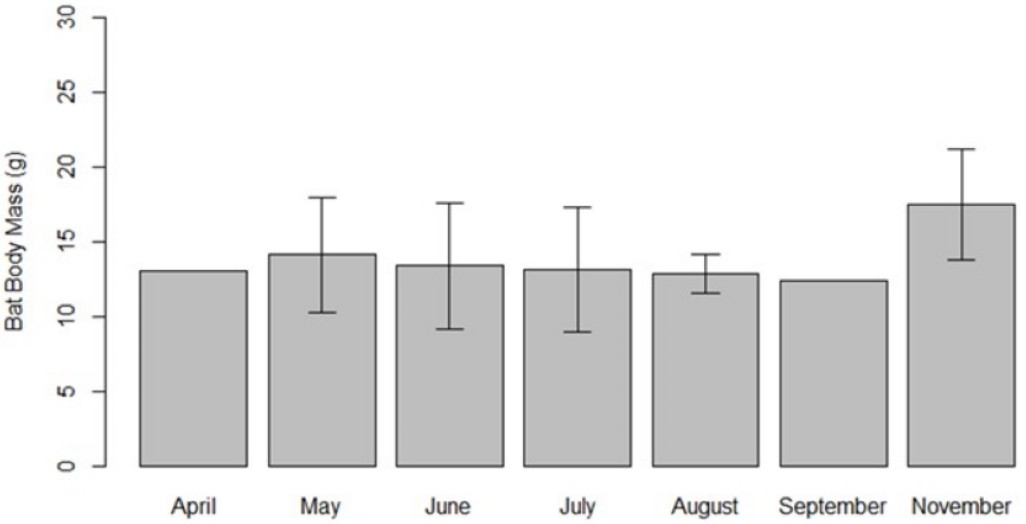

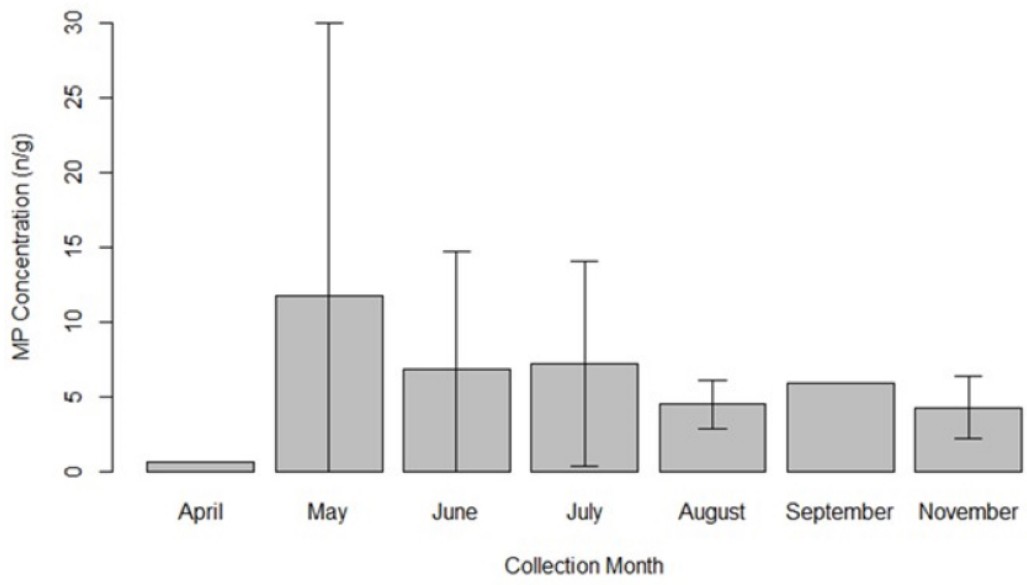

**Figure 4** **Bar graph showing the mean bat body mass (grams) and microplastics concentrations (number of particles per gram of tissue) in gastrointestinal tracts of 26 big brown bats (*Eptesicus fuscus*).** Samples were collected from 15 counties in Tennessee, USA in years 2019 and 2020. The error bars indicate the standard deviation when there was more than one sample. Sample sizes are as follows: April ($n = 1$ bat), May ($n = 8$), June ($n = 4$), July ($n = 5$), August ($n = 3$), September ($n = 1$), and November ($n = 4$).

Studies that are able to collect fresh bat carcasses that have not yet been frozen could assess histopathology of gastrointestinal tracts to determine if bats experience plasticosis

(*Charlton-Howard et al., 2023*). Additionally, studies on the potential role that plasticosis may have in nutrient absorption and fat storage in bats would be highly valuable. If plastics damage cells in GITs, it might make it more difficult for bats to absorb nutrients. For example, celiac disease in humans can compromise the structural integrity of the intestinal lining and inhibit nutrient and vitamin absorption (*García-Manzanares & Lucendo, 2011*).

There is not yet evidence indicating bats selectively consume microplastics based on color. Most of the fibers found in the stomachs, intestines, and lungs of Amazon vampire bats were white or clear (*Alencastre-Santos et al., 2025*). White and clear fibers might be prevalent because many items contributing to microplastic pollution are white or clear before they degrade; however, we are likely also observing the outcome of the degradation of pigments. Over time, colored microplastics are likely to lose pigmentation (*Zhao et al., 2022*). Therefore, less pigmented or unpigmented microplastics will likely always make up some proportion of samples, except where microplastic pollution is recent and pigments have not degraded. Additionally, bleaching during bat food digestion with stomach acid or sample digestion in the lab with the KOH could also be an explanation (*Lee et al., 2022*). While MPs are abundant in the environment, there may be seasonal patterns of bat exposure. For example, bats may not be exposed during hibernation when they limit foraging activities. We were unable to test any seasonal patterns of MP GIT exposure with the limited data; however, we anecdotally noticed that lower MP concentrations may occur during the time immediately post-hibernation and the largest concentrations may occur post-migration and during the early pregnancy period. It is also possible that weather events could alter exposure, as flooding can increase MPs up to 14 times (*Gündoğdu et al., 2018*). This warrants additional research into seasonal patterns of exposure and influence of severe weather events, as bats may be more vulnerable to contaminants and other stressors during periods of high energy expenditure and prey consumption.

It is also possible that not all sympatric insectivorous bat species are exposed to MP equally. It is likely that species traits and geographic factors such as foraging behaviors (*i.e.,* foraging mostly over water sources *versus* terrestrial areas) and proximity to contaminated sites and urban areas would also influence MP exposure. *Eptesicus fuscus* is an urban-adapted bat species, and we received carcasses from programs where the initial source was a direct human-wildlife interaction. Therefore, the bats in this dataset may be more exposed to anthropogenic disturbance and urban development, and, thus, may have higher MP concentrations than bats foraging in less-developed areas. Future studies that use lethal methods or analyze guano from live-captured bats could potentially alleviate some of the bias and achieve a more random sample.

A caveat of this study is that methods in microplastic identification and classification have since evolved beyond the methods used at the time of this study. For example, there are now many suggestions to improve the hot needle method (*Beckingham et al., 2023*). Additionally, Fourier Transformed Infrared Spectroscopy (FTIR) or Raman spectroscopy are the standard now for more sophisticated MP identification and characterization. We had planned to use spectroscopy, but equipment failure led us to use more primitive methods of relying solely on microscopy. Nevertheless, these methods were effective in establishing baseline information.

Our most interesting finding was that higher GIT MP concentrations were associated with bats having lower mass, a proxy for fat reserves (*McGuire et al., 2018*). The model explained 23% of the variation of bat mass with MP concentration; however, we caution that we had a limited sample, and there are likely other factors that influence bat mass. Regardless, our findings indicate that this topic should be explored further. Future studies with larger sample sizes could calculate the limits of detection (LOD) and quantification (LOQ) and explore these and other questions further. Fat storage is a crucial resource for species that migrate and hibernate. Fat storage is even related to disease survival probability, as bats that hibernate and are infected with the non-native pathogen that causes white-nose syndrome are more likely to survive the winter if they start off with higher fat reserves (*Cheng et al., 2019*; *Perry & Jordan, 2020*). Our finding warrants future research exploring this possible relationship between higher MP GIT concentrations and bat mass, on how MPs might block the GIT, inhibit metabolic processes, or the possibility that MP consumption fills the bat with non-nutritional volume and mass that cannot add to body mass and fat stores. The sublethal effects are difficult to study, but could be crucial if MPs remain abundant and persistent contaminants in the environment. Moreover, research on remediation and restoration of aquatic systems and how this might benefit bats and other wildlife would be valuable to future conservation efforts to help recovery of species impacted by emerging environmental contaminants.

## CONCLUSIONS

This study provided evidence that insectivorous bats are exposed to microplastics. Moreover, concentrations in the GIT may influence bat health condition. For example, MP GIT concentrations had a significant influence on bat mass. Future research could investigate the role of MPs in arthropods, drinking water, and particles suspended in the air column in bat ingestion and uptake. Understanding how pollution and environmental contaminants affect bats is critical for monitoring bat populations globally.

## ACKNOWLEDGEMENTS

We acknowledge the help of our lab technicians, Victoria Villanueva, Cheyenne Mireles, and Christopher Fisher. Thank you to Jonathan Reichard for providing the bat stomach content samples. Thanks also to Michael McKinney for sharing his lab space for us to perform the experiments and Richard Gerhold and Eliza Baker for teaching us necropsy techniques.

### Funding

This work was supported by the United States Fish and Wildlife Service (F20AP12217). The funders had no role in study design, data collection and analysis, decision to publish, or preparation of the manuscript.

## Grant Disclosures

The following grant information was disclosed by the authors:
United States Fish and Wildlife Service: F20AP12217.

## Competing Interests

Although Christy Leppanen was an FDA/CTP employee, this work was not done as part of her official duties. This publication reflects the views of the author and should not be construed to reflect the FDA/CTP's views or policies.

## Author Contributions

- Ashleigh B. Cable conceived and designed the experiments, performed the experiments, analyzed the data, prepared figures and/or tables, authored or reviewed drafts of the article, and approved the final draft.
- Emma V. Willcox conceived and designed the experiments, authored or reviewed drafts of the article, and approved the final draft.
- Leah N. Crowley performed the experiments, analyzed the data, authored or reviewed drafts of the article, and approved the final draft.
- Christy Leppanen conceived and designed the experiments, authored or reviewed drafts of the article, and approved the final draft.

## Data Availability

   The raw data for individual bats and the R code used for analysis and figures is available in the Supplemental Files.

## Supplemental Information

Supplemental information for this article can be found online at http://dx.doi.org/10.7717/peerj.19740#supplemental-information.

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
