# Peer review of "First report of microplastics in the gastrointestinal tracts of North American insectivorous bats"

_PeerJ, doi:10.7717/peerj.19740_

## Round 0.1 · original submission · Major Revisions

The three reviewers provide a number of suggestions that should be helpful to the authors in revising their manuscript.

·

Basic reporting

No comment, see attached pdf.

Experimental design

no comment, see attached pdf.

Validity of the findings

No comment, see the attached pdf

·

Basic reporting

no comment

Experimental design

Introduction
Aerial insectivores are among the terrestrial organisms that are exposed to MPs. In the
54 last five years, numerous studies have documented MP exposure in birds (Carlin et al., 2020;55 Fackelmann et al., 2023; Liu et al., 2023; Masiá et al., 2019; Schutten et al., 2024; Teboul et al., 56 2021; Tokunaga et al., 2023; Wayman et al., 2024 tatlı et.all., 2025.)
add this article please;
Tatlı, H. H., Parmaksız, A., Uztemur, A., & Altunışık, A. (2025). Microplastic accumulation in various bird species in Turkey. Environmental Toxicology and Chemistry, 44(2), 386-396.

Write down the habitat and feeding habits of the bat species studied and briefly describe the possibility of encountering microplastics by the insects that are likely to feed on them.


Materyal- metod

Figure 1. add photo of bats in the map

Figure 2ABC. Add ruler showing millimeters.

Include the rationale for not using spectroscopic methods, especially considering that these methods have been more widely adopted in recent studies.

Validity of the findings

this study makes a meaningful contribution to the field of environmental pollution and bat conservation. With minor revisions to enhance the depth of discussion and control for potential lab contamination, this manuscript could be a valuable publication in PeerJ.

Additional comments

The subject of the study is important and has a high potential for original contributions. However, in line with the above criticisms, better presentation of methodological details, hypotheses, and literature connections would make the study scientifically stronger.

Reviewer 3 ·

Basic reporting

The authors present a manuscript detailing a study investigating the presence of microplastics in the stomach contents and GITs of two species of bats, E. fuscus and M. lucifugus. The authors selected a non-cohort of samples from bat carcasses. The samples were not priority for microplastic analysis, such as the sample size, contamination during sample collection, which limited explanation in many issues, such as seasonal pattern of bats exposture to MPs. However, the manuscript showed interesting data, which will be a good basis for future research as the clinical impact for the health of the bats remains unknown.
The paper reads smooth and is strict to the point. The level of English is good, with only a few of typing error (for instance, line 110- incorporate to).

Experimental design

The methodology appears sound and uncomplicated. It is somewhat refreshing to see relatively simple procedures used to generate compelling results in studies such as these. Method described with sufficient detail and information to replicate. Although a techniques hot needle test is almost outdate, the authors show that they are still effective.

Validity of the findings

All underlying data have been provided. Statistically tests are appropriate. Conclusion are well stated and linked to the original research.

Additional comments

I have just some small comments:
The title was edited, please feel free to accept or reject the editing. “First report of microplastics in the stomach contents and gastrointestinal tracts of North America insectivorous bats”
Line 91: the first objective stated that the authors want to test a method for extracting MPs from bats stomach contents stated in line 61. The authors applied the protocol of Microplastic extraction used in GITs for the stomach contents to test a method of extraction and quantification. Please explain why the author use the same protocol. What is modification for stomach contents study, chemicals or condition or both?
Line 39 and 159 : I request you to consult other publications (such as Hale et al. (2020); Vethaak and Legler (2018)) about the size of microplastic should be 1 micrometer - 5 millimeters (5000 micrometers) or 1-5 millimeters.
Line 71: assume, instead of believe.
Line 102-103: Please explain why the author used different pore size of filters for different kind of samples.
Line 156-157 and 159: please correct the number of total samples (574+28 = 602) in line 156-157 or 94% in line 159. I think 574 MP items of 602 items should be 95.35%.
In result,
Sub-header of Method testing with bat stomach contents was expected to indicate the efficiency of the used method. The authors showed only the number and characteristic of MPs.
In discussion,
I expected to see a discussion on technique of microplastic extraction from bat stomach contents and suggestions for the further studies. Again, is the first objective still necessary?
Discuss in the discussion why clear fiber was the most observed MP in GITs. Is clear fiber the most produced MP pollutant by humans? Or do bats incidentally intake via food and water or intentionally select? Can they see colors? Do they think clear fiber is more eatable as it looks like something they recognize? Hypothesize/explain why clear is no 1. Why the result of MP shape and color found in the GITs are different from stomach contents ?
Table 1: the unit of measurement are missing, such as unit of MP count is item or piece, unit of MP concentration is n/g or item/gram, unit of MP size is micrometer.
Table 2 : same comment as table 1.
Figure 2: Scale bars are needed.

---

## Round 0.2 · accepted · Accept

Thanks for your thorough revisions in response to the reviewer feedback.